# Analysis of Movement Variability in Cycling: An Exploratory Study

**DOI:** 10.3390/s23104972

**Published:** 2023-05-22

**Authors:** Lachlan Winter, Clint Bellenger, Paul Grimshaw, Robert George Crowther

**Affiliations:** 1UniSA Allied Health and Human Performance, University of South Australia, Adelaide, SA 5001, Australia; clint.bellenger@unisa.edu.au; 2Alliance for Research in Exercise, Nutrition & Activity (ARENA), University of South Australia, Adelaide, SA 5001, Australia; 3College of Health and Life Sciences, Hamad Bin Khalifa University, Doha P.O. Box 34110, Qatar; pgrimshaw@hbku.edu.qa; 4Faculty of Sciences, Engineering and Technology, Computer and Mathematical Sciences, University of Adelaide, Adelaide, SA 5005, Australia; 5School of Behavioural and Health Sciences, Australian Catholic University, Melbourne, VIC 3065, Australia

**Keywords:** movement variability, Lyapunov Exponent, movement capture, cycling

## Abstract

The purpose of this study was to determine the test-retest repeatability of Blue Trident inertial measurement units (IMUs) and VICON Nexus kinematic modelling in analysing the Lyapunov Exponent (LyE) during a maximal effort 4000 m cycling bout in different body segments/joints. An additional aim was to determine if changes in the LyE existed across a trial. Twelve novice cyclists completed four sessions of cycling; one was a familiarisation session to determine a bike fit and become better accustomed to the time trial position and pacing of a 4000 m effort. IMUs were attached to the head, thorax, pelvis and left and right shanks to analyse segment accelerations, respectively, and reflective markers were attached to the participant to analyse neck, thorax, pelvis, hip, knee and ankle segment/joint angular kinematics, respectively. Both the IMU and VICON Nexus test-retest repeatability ranged from poor to excellent at the different sites. In each session, the head and thorax IMU acceleration LyE increased across the bout, whilst pelvic and shank acceleration remained consistent. Differences across sessions were evident in VICON Nexus segment/joint angular kinematics, but no consistent trend existed. The improved reliability and the ability to identify a consistent trend in performance, combined with their improved portability and reduced cost, advocate for the use of IMUs in analysing movement variability in cycling. However, additional research is required to determine the applicability of analysing movement variability during cycling.

## 1. Introduction

Dynamic systems theory is a concept that explains motor control, suggesting that the neuromuscular system’s organisation is based on the interaction between environmental, organism (i.e., biomechanical and morphological factors) and task constraints [1,2,3]. According to this perspective, movement is non-linear, and changes occur abruptly due to changes in the internal (organism) and/or external (task and environmental) constraints; hence, movement variability is considered normal [3,4]. Variability is considered a feature of healthy function, as it allows the organism to adapt and change their movement patterns to changing constraints. However, there exists a bandwidth of healthy variability. From this perspective, too much variability results in instability, whereas too little variability results in one being too rigid in their movement pattern, rendering the individual unable to properly adapt to changing constraints [4].

There exist various methods to assess movement variability from both linear and non-linear dynamic perspectives [4,5]. A non-linear method used to assess local dynamic stability (the ability to attenuate local perturbations during cyclic movements) [6] is the Lyapunov Exponent (LyE), which measures the divergence of neighbouring trajectories in a state space, the vector area where the dynamical system is defined [7,8]. The larger the variance in trajectories, the more unstable the system, resulting in a larger, positive LyE value, reflecting the idea that one cannot lessen the perturbations in the environment, whereas a locally stable system is indicated by a negative LyE [8,9,10]. The LyE has been useful for understanding the influence of disease [11], ageing [9] and secondary task performance [12] on walking gait mechanics but has seldom been applied to other activities, including cycling.

Cycling is both a popular method of commuting and professional sport; the latter consists of many disciplines, including but not limited to road, BMX and track cycling. Each of these types of cycling has sub disciplines/events. The individual pursuit is an event within track cycling, where males and females cycle for 4000 m and 3000 m, respectively, around a velodrome, with the goal being to complete the event in the shortest time possible [13]. Various factors influence cycling performance, but the primary factors are power output and the resistance experienced by a cyclist, which is primarily rolling resistance and aerodynamic drag, which contribute to 10% and 90%, respectively, of the resistive force a cyclist experiences at typical race speeds [13,14,15]. Cyclists seek to maximise their power output whilst limiting the resistive forces they experience to complete a trial as fast as possible.

In cycling, it can be hypothesised that movement variability should be avoided. Increased movement variability may impact a cyclist’s drag profile by affecting their steering, which would alter the cyclist’s rolling resistance. Additionally, increased movement variability may negatively impact a cyclist’s coefficient of drag, particularly their frontal area, thus impacting their ability to minimise the aerodynamic forces acting against them [15]. Evans et al. [16] determined that longitudinal trunk acceleration (left and right rotation) accounted for 39% of the predicted drag in a 5000 m cycling trial, highlighting that fluctuations in trunk position can impact a cyclist’s drag. This increased movement variability is more likely to be present once the cyclist is fatigued. Additionally, increased movement variability in the lower body and pelvis may alter power production. Lower extremity frontal plane movement shifts will negatively impact the transfer of force onto the pedal, potentially hindering power output [17]. Whilst Wilkinson and Kram [18] concluded that attempting to minimise bicycle lean impacts maximum power output generation negatively, this was concluded in 5 s trials, which are not reflective of any professional cycling distance efforts. As such, quantifying variability during cycling efforts has the potential to be a valuable tool to understand performance. Whilst the LyE will not be able to directly measure a cyclist’s aerodynamic profile or pedal force effectiveness (and its subsequent impact on power output), it will determine the body fluctuations of a cyclist. Theoretically, the lower the LyE value, the more likely the performance will be better, as this indicates that less movement variability is present, which theoretically will reduce aerodynamic drag and maximise power output, maximising performance.

Determining how movement variability is captured and how reliable the data are is critical and must be addressed prior to determining whether variability can provide relevant performance information. Typically, data from optoelectronic measurement systems, such as VICON Nexus, have been utilised to quantify movement variability by calculating segment and joint angular kinematics. However, these systems are expensive, require expertise to operate and are confined to laboratories, rendering in-field assessment implausible. This has resulted in a shift to using inertial measurement units (IMUs). These portable devices consist of a tri-axial accelerometer, gyroscope and magnetometer to measure acceleration, angular velocity and magnetic field strength, as opposed to segment and joint angular kinematics [19,20]. The conversion of acceleration and velocity data to segment and joint angular kinematics can occur with IMUs, but issues arise with the double-integration of data such as sensor drift and offset from the joints’ centres of rotation and the high reliance on sensor orientation accuracy to quantify joint angles, so the accuracy of this method is questionable [21,22,23]. Therefore, analysing the acceleration with IMUs and using the LyE to determine movement variability in real-world settings may potentially be a valuable tool for quantifying performance.

To the authors’ knowledge, little research has utilised non-linear dynamics in cycling and sports in general. Studies utilising the LyE in cycling have assessed variabilities in heart rate [24], pedal time [25] and pedal force [26] with no applications of quantifying movement variability of segment and joint angular kinematics captured from an optoelectronic system such as VICON Nexus and/or segment accelerations captured via IMUs. Despite IMUs having the ability to quantify movement variability in a velodrome setting due to their portability, because of the novelty of this research space, assessments should be first taken in a controlled environment (research laboratory).

Therefore, this study aimed to investigate the test-retest repeatability of Blue Trident IMUs segment acceleration and VICON Nexus segment and joint kinematics in assessing the non-linear dynamics of various body segments or joints in males during three 4000 m cycling sessions in the time trial (TT) position. Secondary analysis will assess differences in variability within the three cycling sessions, breaking the sessions into five intervals. This research seeks to establish if either the Blue Trident IMUs (head, thorax, pelvis and shank accelerations) and/or VICON Nexus segment (thorax, pelvis) and joint (neck, ankles, knees and hips) angular kinematics could repeatedly track movement variability using the LyE across three repeated cycling sessions and across different cycling periods (100 cycling revolutions across 5 interval periods).

It was hypothesised that a moderate to good test-retest repeatability (ICC = 0.6–0.8) would exist between sessions when calculating non-linear dynamics using the LyE across the 100 cycling pedal revolution periods in both IMU segment accelerations and VICON Nexus segment and joint kinematics. It was hypothesised that a significant difference (*p* < 0.05) will exist between the intervals of 100 cycling pedal revolutions, indicating a change in local stability at the assessed sites, therefore demonstrating a greater movement variability as the cycling trial progresses due to fatigue.

## 2. Materials and Methods

### 2.1. Experimental Overview

This study utilised a repeated measures cohort study design. Participants first completed a familiarisation session where the Wattbike Pro (Wattbike, Wattbike Pro, Nottingham, UK) was prepared according to their anthropometric profile. Additionally, they completed a 4000 m ride, which mimicked individual pursuit race distances [13]. During the familiarisation trial, participants could alter the Wattbike resistance to determine their desired resistance for the analysis sessions, which was kept constant to mitigate the effect of extraneous variables. Their chosen resistance was noted to aid with test replication and consistency. Wattbike resistance was controlled by adjusting a lever on the left side of the bike. This alters the air fan wheel vent aperture on the wheel’s centrifugal fan, which alters the Wattbike’s load without altering its measurement system [27,28]. Following the familiarisation session, participants returned for three maximal effort cycling sessions with IMUs and reflective markers attached to their body. To mitigate the effect of fatigue, each cycling session was conducted a minimum of 48 h apart to align with American College of Sports Medicine recommendations and other researchers analysing fatigue giving 48 h of recovery between sessions [29,30,31,32]. Participants were asked to refrain from participating in vigorous exercise prior to each session and altering their normal weekly cycling routines. Additionally, they were asked to maintain a similar diet and fluid (i.e., caffeine, water) intake throughout testing period. To control for diurnal effect, each session was conducted at the same time of day as best as possible [33]. Testing took place at the University of South Australia (UniSA) biomechanics laboratory (City East campus).

### 2.2. Participants

A total of 12 (Mean ± SD, age 25.6 ± 4.5 years, height 1.79 ± 0.07 m, body mass 77.3 ± 9.4 kg) participants were recruited to participate in the study between January and August 2022. All participants who began the testing protocol completed all sessions. All recruited participants were aged between 18–35 years old, had prior experience with maximal intensity exercise and were injury-free on their lower bodies for the previous 6 months to mitigate injury risk and impact of injury on non-linear dynamics results. Additionally, participants must have successfully passed Stage 1 of the Exercise and Sport Science Australia Exercise Pre-Screen system. Ethics approval was obtained from the UniSA Human Research Ethics Committee prior to data collection (protocol number 204077).

### 2.3. Protocol

#### 2.3.1. Informed Consent, Height and Body Mass

Written informed consent was obtained from all participants prior to testing. Upon arrival at the biomechanics laboratory at the UniSA City East campus, height and body mass were measured using a wall-mounted stadiometer and digital scales (TANITA DR-953 Inner Scan Tanita, Tokyo, Japan), respectively.

#### 2.3.2. Bike Fit

To tailor the Wattbike to the individual cyclist, prior to the familiarisation trial, participants underwent a bike fit. If participants already had a tailored bike fit, then they were permitted to replicate this on the Wattbike themselves. If participants did not have a previous bike fit, one was performed by the primary researcher. To tailor the Wattbike to each cyclist, the saddle height was set based on their knee flexion. The saddle height was set at 25 degrees of knee flexion on the right side, measured with a goniometer. The axis of the goniometer was centered on the lateral femoral condyle. Its arms pointed towards the greater trochanter and the lateral malleolus of the ankle. This measurement was taken with the crank at the 6 o’clock position on the right side. Whilst methodological limitations exist with this method, such as the fact that a static fit was conducted for a dynamic movement, it has been shown to optimise performance and mitigate injury risk [34]. To aid with determining this position, participants were fitted with three reflective markers placed at the lateral malleolus, lateral femoral epicondyle and greater trochanter. These markers remained on the participants throughout familiarisation, including during the familiarisation trial, to aid in participants becoming accustomed to wearing them. Similarly, an IMU was placed on the right shank. Methods of determining saddle fore/aft position are much scarcer in the literature, but the most widely applied one is the knee over the pedal spindle method; hence, it was chosen. The saddle fore/aft position was manipulated so that the anterior surface of the patella on the front leg was directly vertical at the pedal spindle when the crank was in a 3 o’clock position [35]. However, due to the lack of biomechanical justifications for this method, this was used as a guideline, so participants were allowed to alter this if they desired in order to improve their comfort. Following this, participants moved into the TT position, where their hands and elbows were placed on the handlebars whilst assuming an almost horizontal trunk position [36]. To standardise the participants’ TT position across all of their sessions, handlebar height, reach and separation were noted [37,38]. Participants wore their own cleats/footwear. The pedals used were E-148 pedals, which were the pedals provided with the Wattbike. For participants who did not own cycling shoes with cleats, a strap accessory was applied so they could attach their feet safely and securely to the pedals. During familiarisation, a Wattbike profile was created for the participant so that cycling performance variables could be collected and later exported for analysis.

#### 2.3.3. Testing Set-Up

Prior to the participants’ arrival, the Wattbike was configured to the participants’ chosen handlebar height, saddle height and fore/aft position and resistance. Additionally, to minimise the impact of reflections from the bike, strapping tape was placed on all metal surfaces.

Prior to testing, 12 VICON MX40+ cameras, capturing at 200 Hz via VICON Nexus, were calibrated using a VICON Nexus wand (VICON, Oxford, UK) moving through the capture volume, per VICON guidelines. The wand set the global origin of the anterior-posterior (A-P; Y) axis of rotation, medio-lateral (M-L; X) axis of rotation and longitudinal axis of rotation (Z) by placing it on the ground within the capture space. Participants wore tight, non-reflective clothing to allow for direct placement of 14 mm reflective markers, which were placed onto bony landmarks of the body with cluster markers placed on the thigh and shank segments. A headband with four markers attached, two on the back and two on the front, was applied to the head. Strapping tape was used to indicate which markers should be at the front and back of the headband. A custom marker set, which was a combination of the CAST and IOR kinematic model marker set, was applied, as demonstrated in Figure 1 [39,40]. Additionally, reflective markers were placed on the Wattbike’s crank and pedal on both the left and right side to aid in determining when a participant completed a pedal revolution. To prevent markers from falling off during sessions, markers were held in place with Elastoplast strapping tape. All markers were applied to the participants prior to them sitting on the Wattbike saddle, apart from the markers attached to the back, which were placed on the participants in the TT position. This was done because the participants’ positions shift greatly as they drop into the TT position. Additionally, the strap accessory rendered it impossible to place markers on the metatarsals of the foot prior to participants sitting on the Wattbike. Hence, markers were placed on the 1st and 5th metatarsal whilst the participant sat on the Wattbike. Due to the design of the strap accessory, the 2nd metatarsal marker was placed between the indent of the strap, roughly in line with the 2nd metatarsal.

Blue Trident IMUs (IMeasureU, Blue Trident, VICON, Oxford, UK) were synchronised with VICON Nexus. They were placed on the head (middle of the forehead), thorax (midline between the bottom of the sternum and the sternum notch), pelvis (between the posterior superior iliac spines) and on the left and right shank (just below the tibial tuberosity), as highlighted with circles in Figure 1 [19,41,42]. IMUs captured segment acceleration at 1125 Hz but were later up-sampled to 1200 Hz within the VICON Nexus software to be a whole number divisor of the Nexus cameras, which ran 200 Hz. Longitudinal, M-L and A-P accelerations were defined as X, Y and Z, respectively. To gauge exercise intensity, heart rate (HR) was monitored with a Polar H10 HR sensor (Polar, Polar H10 Heart Rate Sensor, Polar Australia, Unley, South Australia).

Once participants were fitted with all instruments and were on the Wattbike, a static and dynamic calibration trial was performed. Participants were static for 100 frames in t-pose (standing in the anatomical reference position with arms abducted), and following this, participants completed 10 cycles in the TT position (Figure 2).

Following the calibration trial and marker labelling, the participants’ Wattbike profiles were selected to record the participants’ power output, after which the cycling trial for analysis took place. The medial malleolus marker was removed following calibration to prevent it from colliding with the crank markers (which were required to determine when a pedal revolution occurred).

#### 2.3.4. Cycling Trial

Following being fitted with the reflective markers, IMUs and the HR monitor, participants performed a standardised 5 min warm-up of cycling at a self-selected speed, resistance and cadence with a 5 s maximal intensity cycling effort at the end of the 5 min [43,44,45]. This was followed by a 3 min rest period, after which the cycling trial began.

Participants completed a maximal intensity, 4000 m cycling trial. For the first 200 m, participants were permitted to be out of the saddle but had to assume a seated TT position following 200 m to mimic the push-start in individual pursuit cycling. The distance spent out of the saddle was chosen following private correspondence with an elite cycling coach, who advised that this was the approximate distance spent out of the saddle at the beginning of the race during a push-start where a cyclist will build up speed from a static start.

A piece of yellow tape was placed on the ground 2 m in front of the Wattbike. Throughout the trial, participants were instructed to fixate their gaze on this to mimic the head position that an individual pursuit cyclist assumes whilst cycling. Verbal encouragement was provided to participants to aid them in riding at a maximal intensity. Following completion of the cycling trial, rate of perceived exertion (RPE) was assessed using the 1–10 RPE scale [46].

Upon completion of the cycling trial, participants completed a cool-down on the bike and cycled at a low intensity for at least 3 min. This was done to mitigate the effect of blood pooling and fatigue, given that cycling is a lower-body-dominant movement [47]. Following this, participants were permitted to either complete static stretching of their lower body muscles (hamstrings, quadriceps, lower back and calf muscles) and/or were given a foam-roller to ‘massage’ the lower body.

### 2.4. Data Analysis

#### 2.4.1. Cycling Performance Variables

Mean power output (W), cadence (rpm), pedal force (N) and power to mass ratio (W/kg) were calculated for all participants across both the entire trial and individual periods. These data were collected from the Wattbike and later exported from Wattbike Expert (Wattbike Expert Ver 2.60.20, Colombes, France). Pedal force data are measured by capturing the resultant tangential pedal force through measuring the force on the chain applied through each pedal at a sampling rate of 100 Hz [28].

#### 2.4.2. Cycling IMU Acceleration & Nexus Kinematics

Each cycling trial was separated into 5 intervals, with each interval consisting of 100 cycling pedal revolutions. Based on the fixed gear ratio of the Wattbike, it was estimated that 500 pedal revolutions would be completed in the cycling sessions. However, due to the differing times and resistances each participant rode at, not every participant completed 500 cycles. If 500 cycles were not completed, the number of cycles completed in the last period was reduced and noted. Additionally, the first 200 m of the trial was disregarded due to the participants being out of the TT position.

Gaps in marker trajectories were filled manually using appropriate gap filling options within the VICON Nexus software. C3D files were then exported to Visual3D (v6.0, c-motion, Germantown, MD, USA) for modelling. Within Visual3D, ML axis of rotation (sagittal plane), AP axis of rotation (frontal plane) and longitudinal axis of rotation (transverse plane) movements were defined as flexion/extension, abduction/adduction and rotation movements, respectively. In Visual3D, the trial was reviewed to define two events: the start, when the participant assumed the TT position, and end, where the trial ended. For both IMU and Vicon Nexus data, filtering did not occur. It is recommend that studies calculating the LyE using lower limb joint kinematic data use unfiltered data or filter with a high cut-off frequency [48]. Filtering data may impact the study’s results, as it may result in failing to capture the variability/instability of a system. Afterwards, these segment and joint angular kinematic data were exported for LyE calculation using custom built pipelines.

A custom MATLAB code (The MathWorks, Inc., Natick, MA, USA) was written to calculate the LyE for each interval of each cycling trial for IMU segment acceleration and segment/joint angular kinematic data captured via VICON Nexus. Two separate codes were created for IMU and VICON Nexus data, but both had similar code structures. Briefly, the process is outlined as follows. Using the start point defined in Visual3D, 100 cycles were counted for the 1st interval for both the left and right pedal cycles. Another 4 100-cycle intervals were created; however, instances existed where an additional 100 cycles could not be defined for the 5th interval, so the last pedal cycle was used to define the end point. Whilst previous research analysing the LyE during walking and running kept a consistent number of steps, this was considered the optimal approach, as it meant that additional cycles were not included when participants were out of the TT position, which could impact the LyE result. This that ensured there was a consistent number of cycles across the first four intervals, making comparison easier. In order to calculate LyE, the state space of the dynamical system must be reconstructed [49]. To do this, both the embedding dimension *m* (number of successive points in the dynamical system) and time delay τ (an integer determining how many data points are included for analysis) must be defined. Individual *m* and τ were defined for each variable in each interval. The *m* and time τ were selected using the Global False Nearest Neighbours Method [50] and by determining the first minimum of the Average Mutual Information function [51], respectively, similar to the methods of other researchers applying the LyE [52,53]. Minimum and maximum m and τ are reported in Appendix A. Once the state space was reconstructed, the LyE was calculated using the Rosenstein [54] algorithm, which is the most commonly applied algorithm to calculate LyE in gait research [9]. Following this, the LyE values were exported to Microsoft Excel (Microsoft Corporation, Redmond, WA, USA) for analysis.

### 2.5. Statistical Analysis

Statistical analysis was conducted using SPSS (v28, IBM Corp, New York, NY, USA). Boxplots were used to determine if outliers existed in the data. These data were reviewed to determine if any outliers that were present were not an error. If outliers (even extreme outliers) were present after review, they were still included for analysis. According to DST, movement can be performed in many different ways, so eliminating an outlier and replacing it with the group mean may result in eliminating a ‘real’ result and replacing it with one that does not reflect that individual’s variability. Shapiro and Wilk [55] ‘goodness of fit’ and normality test was used to assessed data distribution. The primary aim was to compare the repeatability of the cycling trial. Using Hopkins’ reliability Microsoft Excel spreadsheet, raw data were analysed [56] and reported as mean ± standard deviation. Test-retest repeatability of VICON Nexus and IMU was assessed via intraclass correlation coefficient (ICC_3,1_), classified according to the following thresholds: <0.50, poor; 0.50–0.75, moderate; 0.75–0.90, good and >0.90, excellent [57]. Standardised typical error of the estimate (TE) was also reported with (thresholds set as small = 0.2, moderate = 0.6, large = 1.2, very large = 2.0 and extremely large = 4.0) [58]. TE of estimate was calculated as two times the observed TE divided by the pure between-subject SD, interpreted using the aforementioned magnitude thresholds [59,60]. In instances where Standardised TE could not be reported; the raw TE was calculated. Instances of this have been noted in the tables. Finally, linear mixed modelling tests were performed to compare interaction between sessions (1 vs. 2 vs. 3) and intervals (1–5) with an alpha set at < 0.05. Due to the number of linear mixed model tests applied, post hoc testing using least squared differences was used to reduce the likelihood of Type 1 error.

## 3. Results

Due to the amount of data, all tables referenced are located within the Appendix A. Table 1 is a summary of the results obtained from this study.

### 3.1. Cycling Performance Variables

All cycling characteristics are reported in Appendix A. Good to excellent reliability (ICC = 0.79–0.97) was found for all cycling Wattbike performance variables, and small to moderate TE was observed (0.44–0.92). No differences were reported in power (W), cadence (RPM) and power to mass ratio (W/kg) between the same intervals across the three sessions (*p* < 0.05). The pedal force in interval 1 increased (*p* < 0.05) in session 1 vs. session 3 (139.3 ± 32.4 N vs. 146.6 ± 30.8 N). Within each session, pedal force (N), power (W) and power to mass ratio (W/kg) decreased (*p* < 0.05) from interval 1 vs. 2, 3, 4 and 5. A similar effect was evident in the cadence (RPM), except in session 1, where a decrease (*p* < 0.05) was observed between interval 1 vs. 2, 3 and 4.

### 3.2. IMU Acceleration LyE

All IMU acceleration LyE variables are presented in Appendix A.

#### 3.2.1. Head

The head acceleration reliability for the LyE is presented in Appendix A. Across the head accelerations, there were moderate to excellent ICCs (0.72 to 0.91), and TE ranged from moderate to large (0.68 to 1.31). The head longitudinal acceleration LyE for interval 1 demonstrated a lower (*p* < 0.05) LyE compared to intervals 3, 4 and 5 in sessions 1 and 3 and from intervals 2–5 in session 2. The head M-L acceleration LyE in sessions 1 and 2 increased (*p* < 0.05) for intervals 1 and 2 vs. intervals 4 and 5 and from interval 3 vs. interval 5. In Session 3, an increase in the LyE occurred from interval 1 vs. intervals 3 to 5, from interval 2 vs. intervals 4 and 5 and from interval 3 vs. interval 5. The head A-P acceleration LyE increased (*p* < 0.05) from interval 1 vs. intervals 3 to 5 in session 1, vs. intervals 4 to 5 in session 2 and interval 5 in session (*p* < 0.05). In sessions 1 and 3, an increase (*p* < 0.05) in the LyE was reported between interval 2 and interval 5.

#### 3.2.2. Thorax

The thorax acceleration reliability for the LyE is presented in Appendix A. Across the thorax accelerations, there were moderate to good ICCs (0.60 to 0.90), and TE ranged from moderate to large (0.71 to 1.33). The longitudinal thorax acceleration LyE increased (*p* < 0.05) from interval 1 vs. intervals 4 and 5 and from interval 3 vs. 5 in all sessions. Similarly, an increase (*p* < 0.05) was reported from interval 2 vs. intervals 4 and 5 and from interval 3 vs. session 5 in both sessions 2 and 3. In session 1, an increase (*p* < 0.05) in the LyE existed from interval 2 vs. 5. In all sessions, the M-L thorax acceleration increased (*p* < 0.05) from intervals 1 and 2 vs. intervals 4 and 5. In sessions 1 and 3, an increase (*p* < 0.05) from interval 3 vs. intervals 4 and 5 and in session 2 from interval 3 vs. 5 occurred. The A-P thorax acceleration LyE increased (*p* < 0.05) in sessions 2 and 3 from interval 1 vs. intervals 4 and 5 and in session 1 to interval 2 vs. 5. In sessions 1 and 3, there was an increase (*p* < 0.05) in the LyE from interval 2 vs. 5 (*p* < 0.05) in the A-P thorax acceleration.

#### 3.2.3. Pelvis

The pelvis acceleration reliability for the LyE is presented in Appendix A. The pelvis acceleration LyE reported poor to good reliability (0.13–0.84) across all three axes, and TE ranged from moderate to large (0.86–1.87). However, improved reliability was reported for longitudinal and the M-L acceleration LyE between sessions 2 and 3, indicating a potential learning effect. The M-L pelvis acceleration LyE increased (*p* < 0.05) from session 3 between intervals 2 vs. 5. The longitudinal pelvis acceleration LyE decreased (*p* < 0.05) in intervals 1 and 2 from session 1 vs. 3.

#### 3.2.4. Shanks

The left shank (LS) and right shank (RS) acceleration reliability for the LyE is presented in Appendix A. Across all three axes, the LS (0.38–0.78) and RS (0.15–0.81) shank acceleration LyE had poor to good ICCs, and TE ranged from moderate to large (1.02 to 1.61) (0.94–1.86), respectively. However, improved reliability was reported for the A-P RS acceleration LyE between sessions 2 vs. 3, indicating a potential learning effect. The longitudinal LS acceleration LyE increased (*p* < 0.05) from session 1 vs. 2 in interval 3 and in interval 4 increased (*p* < 0.05) from session 1 vs. sessions 2 and 3. The M-L lateral LS acceleration LyE increased (*p* < 0.05) from session 1 vs. 2 in interval 3 and increased (*p* < 0.05) in intervals 4 and 5 from session 1 vs. sessions 2 and 3. The A-P LS acceleration LyE increased (*p* < 0.05) from session 1 to both sessions 2 and 3 in intervals 2, 3 and 5. In interval 4, the A-P LS acceleration LyE increased (*p* < 0.05) from session 1 vs. 2. The longitudinal RS acceleration LyE increased (*p* < 0.05) from session 1 vs. 2 in intervals 2 to 4. The M-L RS acceleration LyE increased (*p* < 0.05) from session 1 vs. 3 in both intervals 4 and 5.

### 3.3. VICON Nexus Segment and Joint Angular Kinematics

All VICON Nexus segment and joint angular kinematics LyE variables are presented in Appendix A.

#### 3.3.1. Neck

The neck kinematic reliability for the LyE is presented in Appendix A. Across the neck kinematic LyEs, there were poor to excellent ICCs (0.18 to 0.99), and TE ranged from small to large (0.18 to 1.76). The neck flexion/extension LyE increased (*p* < 0.05) from interval 3 vs. 5 in both sessions 1 and 3. The neck lateral flexion LyE in sessions 1 and 2 increased (*p* < 0.05) from interval 1 vs. 5 and from interval 2 vs. 5, respectively. Similarly, the neck rotation LyE increased (*p* < 0.05) from interval 1 vs. intervals 3 and 4 in session 1 and from interval 1 vs. intervals 2 to 5 in session 2, respectively. The neck rotation LyE decreased (*p* < 0.05) in interval 2 from session 2 vs. 3.

#### 3.3.2. Thorax

The thorax kinematic reliability for the LyE is presented in Appendix A. Across the thorax segment kinematic LyEs, there were poor to moderate ICCs (0.03 to 0.54), and TE ranged from moderate to very large (1.05 to 2.03). The spinal lateral flexion LyE increased (*p* < 0.05) in intervals 1 and 4 from session 1 vs. 2 and in interval 5 from session 1 vs. 3. An increase (*p* < 0.05) in the spinal lateral flexion LyE was reported from interval 1 vs. 5 in session 1 and from interval 2 vs. 5, in session 2, respectively.

#### 3.3.3. Pelvis

The pelvis kinematic reliability for the LyE is presented in Appendix A. Across the pelvis kinematic LyEs, there were poor to moderate ICCs (0.00 to 0.70), and TE ranged from moderate to very large (1.16 to 2.00). The pelvic tilt LyE decreased (*p* < 0.05) from interval 1 vs. 3 in session 1 and from interval 3 vs. 4 in session 3. Similarly, a decrease (*p* < 0.05) in the pelvis obliquity LyE was reported from interval 1 vs. 5 in session 1. The pelvic tilt LyE decreased (*p* < 0.05) in interval 3 from session 1 vs. 3, and pelvic obliquity decreased (*p* < 0.05) from session 1 vs. sessions 2 and 3 for interval 1.

#### 3.3.4. Hip

The left (LH) and right hip (RH) kinematic reliability for the LyE is presented in Appendix A. Across the LH (0.37 to 0.74) and RH (−0.04 to 0.68) hip kinematics, there were poor to moderate ICCs. LH and RH kinematic LyE TE ranged from moderate to large (1.10 to 1.63) and large to very large (1.31 to 2.04), respectively. A decrease (*p* < 0.05) in the LH rotation angle LyE was reported from interval 1 vs. 3 in session 2. In session 3, a decrease (*p* < 0.05) in the LH rotation LyE was reported from interval 1 vs. 3 and increased (*p* < 0.05) in LyE from interval 3 vs. intervals 4 and 5. In session 1, the RH abduction/adduction LyE increased (*p* < 0.05) from intervals 2 and 3 vs. interval 5 and in session 3 from interval 2 vs. 4 (*p* < 0.05). The RH abduction/adduction LyE increased (*p* < 0.05) in intervals 2 and 3 from session 1 vs. 2, in intervals 3 and 4 from session 1 vs. 3 and in interval 4 from session 2 vs. 3. A decrease in RH rotation was reported from session 1 vs. 2 in interval 5 (*p* < 0.05).

#### 3.3.5. Knee

The left (LK) and right knee (RK) kinematic reliability for the LyE is presented in Appendix A. Across the LK (0.21 to 0.76) and RK (0.33 to 0.81) kinematics, there were poor to good ICCs. The LK (1.06 to 1.73) and RK (0.95 to 1.78) kinematic LyE TE ranged from moderate to large, respectively. In session 1, an increase in the LK flexion/extension LyE was reported from interval 2 vs. interval 5 (*p* < 0.05). The LK rotation LyE in session 1 decreased (*p* < 0.05) from interval 1 vs. intervals 3 to 5 and from interval 2 vs. 3. Similarly, in session 2, a decrease (*p* < 0.05) in LK rotation from interval 1 vs. intervals 3 and 5 and from interval 2 vs. 5 was reported, and in session 3, a decrease from intervals 1 and 2 vs. interval 3 occurred. In session 1, a decrease (*p* < 0.05) in the RK rotation LyE was reported from interval 1 vs. 4. The LK flexion/extension LyE decreased (*p* < 0.05) in interval 5 from session 1 vs. 2 and session 2 vs. 3.

#### 3.3.6. Ankle

The left ankle (LA) and right ankle (RA) kinematic reliability for the LyE is presented in Appendix A. Across the LA (0.22 to 0.76) and RA (0.20 to 0.78) kinematics, there were poor to good ICCs. The LA (1.04 to 1.79) and RA (1.02 to 1.81) kinematic LyE TE ranged from moderate to large, respectively. In interval 1, a decrease (*p* < 0.05) in the LyE was reported from session 1 vs. 2 and from session 1 vs. 3 in the left foot rotation. No between-session differences were reported for ankle dorsi/plantarflexion, foot progression and right foot rotation.

## 4. Discussion

This study aimed to test the test-retest repeatability of both Blue Trident IMUs and VICON Nexus kinematic modelling in assessing movement variability and to determine if variability changes existed at different segments/joints during a 4000 m cycling bout. No study, to the authors’ knowledge, has assessed kinematic variability using the LyE during cycling. Therefore, first determining if the applied methods are repeatable prior to application in competition and training scenarios is necessary. It was hypothesised that a moderate to good test-retest repeatability and significant differences would exist within sessions in the LyE for the IMU and VICON Nexus segment and joint angular kinematics. The cycling protocol implemented had excellent test-retest repeatably for cycling performance variables, providing a strong base to review changes in IMU and VICON Nexus variability. Predominately, poor to good and poor to moderate/good reliability was reported for IMU and VICON Nexus kinematic modelling, respectively. However, the IMU acceleration demonstrated a consistent trend, whereby head and thorax acceleration increased across the bout, with no alteration in pelvic and shank acceleration. A consistent trend did not exist in VICON Nexus variability.

### 4.1. Cycling Performance

This study used a new design to assess movement variability. Based on the results of the cycling performance test-retest repeatability, the protocol implemented was designed and executed well. It was expected that across a bout, fatigue would occur and influence performance. In each session, power, power to mass ratio, cadence and pedal force peaked in interval 1 and decreased across the bout, with a slight increase in these variables from intervals 4 to 5, thereby demonstrating potential fatigue. The protocol implemented was reliable, and therefore, splitting variability results into intervals is a valid method to understand variability across a cycling bout.

### 4.2. Upper Body Variability

The longitudinal, M-L and A-P head and thorax acceleration LyE increased across the bout; thus, greater instability occurred as the participant fatigued. The observed changes are reliable, as the head and thorax acceleration LyE reliability was moderate to excellent and moderate to good, respectively. Similarly, VICON Nexus neck and thorax kinematic variability increased across the session but to a lesser extent as a consistent increase in variability was not experienced throughout the bout, except in neck rotation in session 2. Reliability and TE in LyE thorax kinematics was decreased in VICON Nexus kinematic modelling compared to IMU acceleration, likely due to the TT position causing the participants’ upper arms and handlebars to occlude the xiphoid process and manubrium markers, resulting in large ‘gaps’ in these marker trajectories, an issue not present when capturing thorax segment application with an IMU. Increased thorax acceleration may have been adopted to maintain power output by shifting more body weight to the pedals [61]. Alterations in head and thorax acceleration and therefore position, particularly along the frontal plane (Z axis of rotation [anterior-posterior] for IMUs) will alter a cyclist’s frontal area, increased their aerodynamic drag and consequently, negatively impacted performance [62,63]. However, to validate this, correlations between increased head and thorax variability and increased drag must be made in wind-tunnel tests. The impact of an increased frontal area is more pronounced in elite cyclists, where minimising drag is essential. Examining variability in elite cyclists and comparing that to novices will provide insight into the effect of increased head and thorax acceleration on performance. Theoretically, elite cyclists should be able to better minimise thorax acceleration due to their experience in the TT position, but further investigation is required to confirm this. Additionally, lateral trunk movements should be avoided due to their association with increased incidences of lower back pain in cycling [64].

### 4.3. Lower Body Variability

Whilst increased IMU acceleration variability was reported across the bout at the head and thorax, this did not occur at the pelvis. However, pelvis acceleration variability differed across separate cycling sessions. VICON Nexus kinematic variability reported different changes in the pelvis LyE than IMU acceleration. IMUs reported instability at the pelvis, indicated by the positive LyE and VICON Nexus segment/joint kinematic modelling reported stability, indicated by the negative LyE. This highlights that the results obtained from each analysis method cannot be compared because they measure different quantities. Therefore, to better understand the differences in the abilities of IMU and VICON Nexus kinematic modelling in analysing the LyE, examining the same variable is required. Between-session differences were also present in IMU shank acceleration, potentially due to the orientation of the tibia, making it more subject to skin and flesh artefacts [65]. However, IMUs were placed just below the tibial tuberosity to best ensure placement consistency [19]. VICON Nexus hip and knee kinematic variability had no consistent pattern in variability changes, both within and between sessions, and limited alterations occurred in ankle variability.

### 4.4. IMUs vs. Kinematic Modelling

A broader range of variability reliability was reported for VICON Nexus kinematic modelling compared to IMU acceleration. Furthermore, better TE was generally reported for IMU acceleration than VICON Nexus kinematic modelling. This may have occurred due to the complexity of the reflective marker model, which required 38 reflective markers with five four-marker clusters. Slight changes in marker position could alter the joint/segment kinematic results, impacting the LyE and, therefore, the reliability. Conversely, IMUs have less potential for error due to having only five sites analysed and segment acceleration data being less effected by consistency in sensor placement. Therefore, due to their improved reliability and TE, their significantly reduced cost, both in expense and set-up time, and their improved portability, it is recommended that IMUs are selected for analysing the LyE rather than kinematic modelling via motion capture.

### 4.5. Limitations

The results’ lack of generalisability is a limitation. The recruited participants were all novice cyclists without prior experience in TT position cycling. Although good to excellent ICCs in cycling performance variables were demonstrated, the differences in variability across the session could have occurred due to participants adopting different movement patterns as they became more accustomed to the TT position. Comparisons to and assessments of elite cyclists are required because elite cyclists should now be the future focus. Furthermore, to move closer to understanding environmental influences, calculating the LyE whilst cycling is being performed on a velodrome with a mobile bicycle rather than a Wattbike in a research laboratory is necessary. Finally, without a direct assessment of variability to aerodynamic drag, inferences can only be made about the potential effect of increased variability on drag and, subsequently, performance.

## 5. Conclusions

To the authors’ knowledge, this study was the first that has examined the test-retest reliability for assessing non-linear movement variability during a cycling bout using two different methods of movement capture (IMUs and VICON Nexus). Inconsistent trends were reported amongst VICON Nexus segment/joint kinematic modelling. The IMUs’ acceleration variability analysis demonstrated that participants increased their head and thorax variability across a bout without altering pelvis and shank variability, which may have had detrimental effects on aerodynamic drag. Given the outcome of this study and IMUs’ portability and reduced time and money costs, their use is recommended over VICON Nexus kinematic modelling. Future research should investigate the relationship between changes in movement variability and the relationship to drag profiles prior to application in competition and training scenarios.

## Figures and Tables

**Figure 1 sensors-23-04972-f001:**
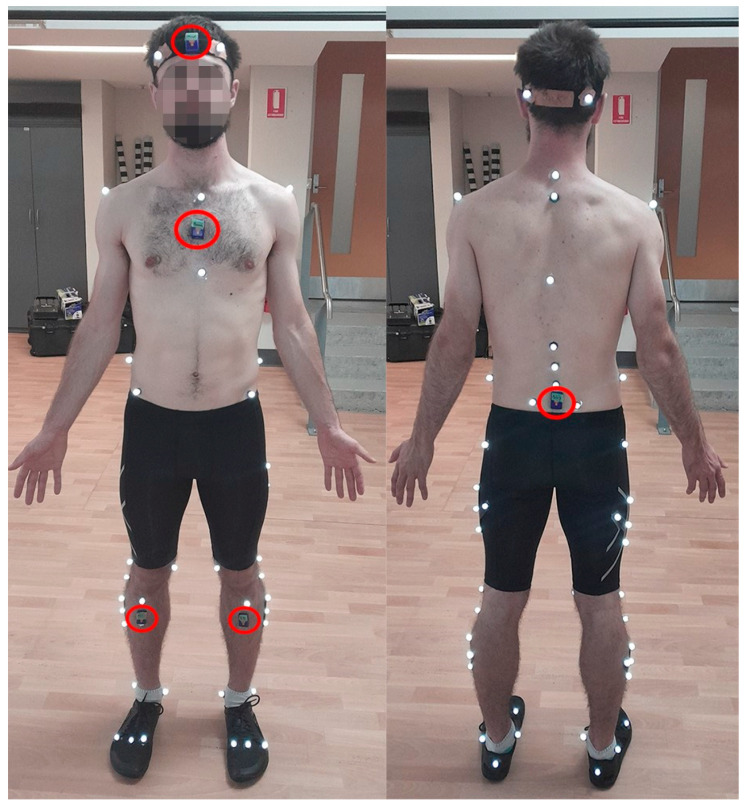
Placement of reflective markers and IMUs on the participant.

**Figure 2 sensors-23-04972-f002:**
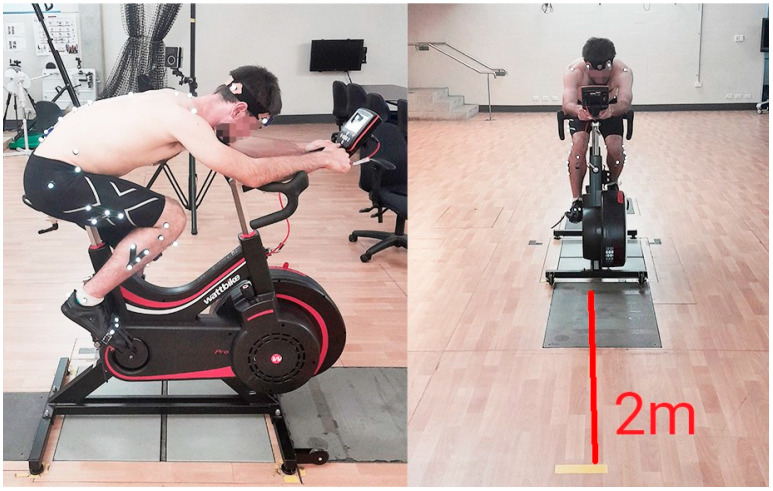
Sagittal and frontal view of the participant in the TT position.

**Table 1 sensors-23-04972-t001:** Summary of results.

	Result
IMU	
Head Acceleration	Moderate to excellent ICC, moderate to large TE. LyE decreased across the bout in all three axes, in each session, with no change between intervals of the same session.
Thorax Acceleration	Moderate to good ICC, moderate to large TE. LyE decreased across the bout in all three axes, in each session, with no change between intervals of the same session.
Pelvis Acceleration	Poor to good ICC, moderate to large TE. No consistent trend in LyE changes.
Shank Acceleration	Poor to good ICC, moderate to large TE for both LS and RS. LS and RS acceleration reported differences between sessions in all 3 axes, with no within session differences.
VICON NEXUS	
Neck Kinematics	Poor to excellent ICC, small to large TE. Neck rotation LyE decreased in interval 2 from session 2 to 3, no other between-session differences occurred. Alterations to LyE within session occurred in all 3 planes.
Thorax Kinematics	Poor to moderate ICC, moderate to very large. TE Spinal lateral flexion LyE increased within the session.
Pelvis Kinematics	Poor to moderate ICC, moderate to very large TE. Decreased pelvic tilt and obliquity LyE in session 1 and 3 within and between sessions, with no consistent trend.
Hip Kinematics	Poor to moderate ICC for LH and RH, moderate to large TE for the LH, large to very large TE for the RH. LH rotation differences within session, with no consistent trend. RH abduction/adduction reported between and within session differences and RH rotation decreased from session 1 to 2 in interval 5.
Knee Kinematics	Poor to good ICC, moderate to large TE for both LK and RK. LK rotation decreased across the bout within all sessions and RK decreased from interval 1 to 4 in session 1. Limited within session differences for knee flexion/extension but LK flexion/extension variability reduced from session 1 to 2 and 2 to 3, indicating a learning effect.
Ankle Kinematics	Poor to Good ICC and moderate to large TE for both LA and RA LyE. No differences between or within session bare left rotation decreasing from session 1 to 2 and session 1 to 3 in interval 1.

## Data Availability

Data are available on request to the corresponding author.

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
