# Peer review of "Analysis of Movement Variability in Cycling: An Exploratory Study"

_sensors, 2023, doi:10.3390/s23104972_

Round 1
Reviewer 2 Report
The manuscript presents a study aimed at testing the repeatability of inertial measurement units (IMUs) and a motion capture system (VICON Nexus) in analyzing variability during cycling. The experiment involved cyclists who completed four 4000m cycling trials with inertial sensors and markers attached to various body parts. During cycling, IMU accelerations at the head and chest increased, while those at the pelvis and shank remained consistent. Motion capture data showed differences across trials but no clear pattern. IMUs demonstrated better repeatability, detected consistent trends, and were more portable and affordable.
The research is novel because it employs the Lyapunov exponent approach, which has previously been used mainly to study walking, and less frequently for cycling. A lower Lyapunov exponent may indicate better performance due to less variability, resulting in reduced drag and increased power.
The authors acknowledge the study's limitations, including a lack of generalizability and elite athlete representation. The novice participants were unfamiliar with the time trial position and may have changed their movement patterns during the sessions, impacting variability. The authors suggest that future research should focus on elite cyclists, examine variability in velodrome settings, directly measure aerodynamic drag, and assess its impact on performance.
Line 19, familirisation, spelling, also in lines 135, 167, 180, 194, and possibly elsewhere, please check
Line 16, please add abbreviation (IMU), or spell it out in line 20
Line 60, I would change "quickest" to "shortest"
Line 72, I do not understand how longitudinal trunk acceleration could influence drag, by what mechanism. Please consider adding an explanation.
Line 156, please remove "within"
Line 167, capital W in Wattbike, at some other locations as well
Lines 239-240, what is "t-pose"? Please consider adding an explanation.
Lines 90-92, The sentence was difficult for me to follow. Please consider rewriting it, possibly using more sentences, for example like this: "Optoelectronic measurement systems, such as VICON Nexus, are typically used to quantify movement variability. They do this by calculating the kinematics of body segments and joints, such as angles."
Lines 94-103, This text reads somewhat selfcontradictory. The contradiction seems to be that optoelectronic systems are said to quantify movement variability by calculating joint kinematics, but then IMUs are suggested as an alternative precisely because calculating joint kinematics from IMUs can be inaccurate. Please consider rephrasing.
Lines 170-174, I had hard time to understand this sentence. It could perhaps become clearer if split into shorter ones, for example like this: "To tailor the Wattbike to each cyclist, the saddle height was set to match their knee flexion. The saddle height was set at 25 degrees of knee flexion on the right side, measured with a goniometer. The axis of the goniometer was centered on the lateral femoral condyle. Its arms pointed towards the greater trochanter and the lateral malleolus of the ankle. This measurement was taken with the crank at the 6 o’clock position on the right side."
Lines 192-193, another sentence I would suggest to rephrase for clarity, for example like this: "For participants who did not own cycling shoes with cleats, a strap accessory was applied so they could attach their feet safely and securely to the pedals."
Lines 294-295, there seems to be an issue with grammar in this sentence. Please consider rephrasing, for example like this: "It is recommended studies calculating the LyE from lower limb joint kinematic data use unfiltered or high cut-off filtered data."
Lines 559-560, again, a sentence difficult for me to follow. Please consider rephrasing, perhaps like this: "During cycling, the acceleration of the head and chest area steadily increased, as measured by an motion sensor. However, the acceleration of the pelvis region did not increase similarly within the same session. That said, the pelvis acceleration differed across separate cycling sessions."
Reviewer 3 Report
This reviewer appreciates the time and efforts invested by the authors in reporting their investigation. Th overall manuscript is ell written. This reviewer hand included the comments at requisite portions in the attached pdf file.
